# IQGAP2 Inhibits Migration and Invasion of Gastric Cancer Cells via Elevating SHIP2 Phosphatase Activity

**DOI:** 10.3390/ijms21061968

**Published:** 2020-03-13

**Authors:** Liang Xu, Yuling Shao, Lin Ren, Xiansheng Liu, Yunyun Li, Jiegou Xu, Yan Ye

**Affiliations:** Department of Immunology, School of Basic Medical Sciences, Anhui Medical University, Hefei 230032, China; LiangXu2019@163.com (L.X.); 15955118431@163.com (Y.S.); renlin2222@163.com (L.R.); xianshengliu12138@163.com (X.L.); liyunyunxd123123@163.com (Y.L.); xujiegou@ahmu.edu.cn (J.X.)

**Keywords:** gastric cancer, IQGAP2, SHIP2, migration, invasion

## Abstract

Previous studies have shown reduced expression of Src homology 2-containing inositol 5-phosphatase 2 (SHIP2) and its tumor-suppressive role in gastric cancer (GC). However, the precise role of SHIP2 in the migration and invasion of GC cells remains unclear. Here, an IQ motif containing the GTPase-activating protein 2 (IQGAP2) as a SHIP2 binding partner, was screened and identified by co-immunoprecipitation and mass spectrometry studies. While IQGAP2 ubiquitously expressed in GC cells, IQGAP2 and SHIP2 co-localized in the cytoplasm of GC cells, and this physical association was confirmed by the binding of IQGAP2 to PRD and SAM domains of SHIP2. The knockdown of either SHIP2 or IQGAP2 promoted cell migration and invasion by inhibiting SHIP2 phosphatase activity, activating Akt and subsequently increasing epithelial–mesenchymal transition (EMT). Furthermore, knockdown of IQGAP2 in SHIP2-overexpressing GC cells reversed the inhibition of cell migration and invasion by SHIP2 induction, which was associated with the suppression of elevated SHIP2 phosphatase activity. Moreover, the deletion of PRD and SAM domains of SHIP2 abrogated the interaction and restored cell migration and invasion. Collectively, these results indicate that IQGAP2 interacts with SHIP2, leading to the increment of SHIP2 phosphatase activity, and thereby inhibiting the migration and invasion of GC cells via the inactivation of Akt and reduction in EMT.

## 1. Introduction

The aberrant activation of phosphatidylinositol-3 kinase (PI3K)/ protein kinase B (PKB, also known as Akt) signaling, is considered to promote tumor development and progression [1]. PI3K activation is initiated by the engagement of receptor tyrosine kinases (RTKs) to extracellular growth factors, which recruits PI3K to plasma membrane-anchored receptors and results in the conversion of PI3K phosphorylates phosphatidylinositol(4)bisphosphate (PI(4)P) and phosphatidylinositol(4,5)bisphosphate (PI(4,5)P2) to phosphatidylinositol(3,4)bisphosphate (PI(3,4)P2) and phosphatidylinositol(3,4,5)trisphosphate (PI(3,4,5)P3). PI(3,4)P2 and PI(3,4,5)P3 subsequently bind to and activate multiple downstream effectors [2]. Among them, Akt is an important effector protein kinase which is highly activated in nearly 80% of gastric cancers (GCs), and its activation may serve as a biomarker for the diagnosis of GC and as a molecular target for treatment [3]. It has been reported that some lipid phosphatases, such as phosphatase and tensin homolog deleted on chromosome 10 (PTEN), inositol polyphosphate 4-phosphatase type I (INPP4A) and type II (INPP4B), acting as tumor suppressors, negatively regulate the PI3K/Akt signaling via the dephosphorylation of PI(3,4)P2 and PI(3,4,5)P3 at the 3- and 4-position [4,5,6]. However, the inhibitory effect of phosphoinositide 5-phosphatases on PI3K/Akt signaling remains controversial. One reason is that their product, PI(3,4)P2, is also a positive regulator of Akt activation, although it can be further hydrolyzed by 4-phosphatases; another is that the N- and C-terminal domains of 5-phosphatases influence their phosphatase activities by interacting with various proteins [7]. Nevertheless, it has been demonstrated that 5-phosphatases, such as Src homology 2-containing inositol 5-phosphatase (SHIP) in hematopoietic cells [8], Phosphatidylinositol 4,5-Bisphosphate 5-Phosphatase (PIB5PA) in neuritis and melanoma cells [9,10], and Src homology 2-containing inositol 5-phosphatase 2 (SHIP2) in glioblastoma and GC cells, inhibit Akt activation [11,12].

SHIP2, encoded by inositol polyphosphate phosphatase-like 1 (INPPL1), belongs to the phosphoinositide 5-phosphatase family, which has been implicated in some human diseases such as type 2 diabetes, Alzheimer’s disease and Opsismodysplasia [13,14,15]. However, the role of SHIP2 in tumorigenesis and tumor progression remains paradoxical: high SHIP2 expression has been found in breast cancer, hepatocellular cancer, non-small cell lung cancer, and colorectal cancer, which correlates with poor survival of patients and contributes to the malignant potential of these tumors [16,17,18,19], whereas reduced expression of SHIP2 in GC promotes tumorigenesis and proliferation of GC via activation of the PI3K/Akt signaling [12], and, in squamous cell carcinoma and glioblastoma cells, SHIP2 inhibits Akt activation and leads to apoptosis and cell cycle arrest [11,20]. The current evidence suggests that the pro- or anti-tumorigenic effect of SHIP2 largely depends on cell context, and SHIP2 multi-functional domains may account for its contradictory roles in different cancer cells. Besides the central 5-phosphatase catalytic domain, SHIP2 possesses an N-terminal SH2 domain, a C-terminal proline-rich domain (PRD), and a unique sterile alpha motif (SAM) domain, which affect a variety of biological functions, including cell adhesion, migration, invasion and receptor internalization [21,22,23]. Although our previous work indicated the reduced expression of SHIP2 and its tumor-suppressive role in GC [12], the precise role of SHIP2 in the migration and invasion of GC cells remains to be delineated. By immunoprecipitation and mass spectrometry assays, we identified a potential partner of SHIP2, IQ motif-containing GTPase-activating protein 2 (IQGAP2), which is a highly conserved scaffolding protein that plays a role in cytoskeleton regulation by juxtaposing Rho GTPase and Ca^2+^/calmodulin signals [24]. Loss or down-regulation of IQGAP2 has been found in hepatocellular carcinoma, prostate cancer, ovarian cancer and GC [25,26,27,28]. Moreover, cumulative evidence from clinical specimens supports that IQGAP2 functions as a potential tumor suppressor [29]. While the roles and underlying mechanisms of IQGAP2 in hepatocellular carcinoma, prostate cancer and ovarian cancer have been demonstrated [25,26,27], the biological functions of IQGAP2 in GC are still elusive.

Here, we show for the first time that IQGAP2 interacts with SHIP2 and enhances its phosphatase activity, thereby inactivating Akt and reducing epithelial–mesenchymal transition (EMT), which as a result inhibits the migration and invasion of GC cells. This study thus describes the roles of IQGAP2 and SHIP2 in the migration and invasion of GC cells and provides new insights into the mechanisms of GC progression.

## 2. Results

### 2.1. Expression of IQGAP2 and SHIP2 in Human GC

As reduced IQGAP2 expression was frequently observed in several cancers which correlated with poor survival, we examined the expression of IQGAP2 in 9 GC cell lines and the human normal gastric mucosal epithelial cell line GES-1 by Western blot and qRT-PCR analyses. While SHIP2 expression was consistently lower in GC cell lines than that in GES-1, IQGAP2 was generally expressed in GES-1 and GC cell lines, except MKN-28 and NCI-N87 cell lines, at both mRNA and protein levels (Appendix A). To determine the expression of IQGAP2 in GC tissues, we analyzed The Cancer Genome Atlas (TCGA) database using UALCAN platform (http://ualcan.path.uab.edu) and Human Protein Atlas (HPA) database (http://www.proteinatlas.org). Compared with normal tissues, there was no significant difference in IQGAP2 expression in stomach adenocarcinoma specimens (Appendix A).

### 2.2. SHIP2 Physically Interacts with IQGAP2 through PRD and SAM Domains

To validate endogenous interaction between SHIP2 and IQGAP2, we performed co-immunoprecipitation (Co-IP) assays with the lysates of two GC cell lines co-expressing IQGAP2 and SHIP2 (SGC-7901 and MGC-803). As in the results shown in Figure 1A,B, IQGAP2 and SHIP2 can reciprocally immunoprecipitate each other with anti-SHIP2 or anti-IQGAP2 antibodies. Immunofluorescence staining showed that SHIP2 and IQGAP2 co-localized mainly in the cytoplasm of GC cells (Figure 1C). These results confirmed the endogenous interaction between SHIP2 and IQGAP2 in GC cells.

To unveil which domains of SHIP2 are responsible for this interaction, we made wild-type and a series of deletion mutant constructs of SHIP2 (Figure 2A), and then co-transfected these constructs and cMyc-IQGAP2 plasmids into HEK293T cells. Co-IP assays confirmed the interaction between IQGAP2 and SHIP2 (Figure 2B,C). Additionally, the results showed that SHIP2 truncates without PRD and C-terminal region (including the SAM domain) could not interact with IQGAP2. However, the absence of SH2 and 5-Phosphatase catalytic domains did not affect interaction between them (Figure 2B,C). These data indicated that PRD and SAM domains of SHIP2 are crucial for the binding of IQGAP2 to SHIP2.

### 2.3. IQGAP2 Inhibits Migration and Invasion of GC Cells Correlated with SHIP2 Enzyme Activity

To determine whether the interaction between SHIP2 and IQGAP2 affects their expression, we knocked down SHIP2 and IQGAP2 in SGC-7901 and MGC-803 cells by infecting lentivirus particles carrying SHIP2 shRNA or IQGAP2 shRNA. Western blot analysis showed that knockdown of either SHIP2 or IQGAP2 did not affect the expression of each other (Figure 3A). However, knockdown of either SHIP2 or IQGAP2 increased Akt activation, and induced up-regulation of N-Cadherin and down-regulation of E-Cadherin and β-Catenin, markers for EMT (Figure 3A). Therefore, we performed malachite green phosphate assays to investigate whether IQGAP2 affects SHIP2 enzyme activity, and found that 5-phosphatase activity of SHIP2 was reduced by the knockdown of IQGAP2, similar to the effect of SHIP2 knockdown (Figure 3B). These results revealed that interaction between SHIP2 and IQGAP2 may affect the 5-phosphatase activity of SHIP2.

Then, to further explore the effect of IQGAP2 knockdown on GC cell migration and invasion, we did wound-healing assays and transwell assays, and found that IQGAP2 knockdown promoted cell migration and invasion in SGC-7901 and MGC-803 similar to the effect of SHIP2 knockdown (Figure 3C,D). Collectively, the results suggested that IQGAP2 inhibited the migration and invasion of GC cells, which may be associated with elevating SHIP2 enzyme activity and thereby inactivating Akt and reducing EMT.

### 2.4. The Inhibition of IQGAP2 on Migration and Invasion of GC Cells Is Partially Due to Elevating SHIP2 Phosphatase Activity

Next, to clarify the mechanisms of IQGAP2 in GC cells’ migration and invasion, we knocked down IQGAP2 in stable ectopic SHIP2-overexpressing GC cells (SGC-7901.SHIP2). Wound-healing and transwell assays revealed that the knockdown of IQGAP2 significantly reversed the inhibitory effect of SHIP2 overexpression on cell migration and invasion, which was associated with suppression of elevated SHIP2 phosphatase activity and activation of Akt (Figure 4A–E). These data may be partially attributed to IQGAP2 directly binding to SHIP2 and increasing its activity, and therefore inhibiting the Akt-EMT signaling pathway.

In a further study, to address whether the binding of IQGAP2 to PRD and SAM domains of SHIP2 participated in the inhibitory effects of IQGAP2 on the migration and invasion of GC cells, we introduced wild-type SHIP2 or deletion mutant SHIP2△935-1258 plasmids into SGC-7901.sh-SHIP2 stable cell lines. As shown in Figure 5A,B, the introduction of wild-type SHIP2 plasmids enhanced phosphatase activity of SHIP2, inhibited Akt activation, up-regulated E-Cadherin and β-Catenin, and down-regulated N-Cadherin compared with that of deletion mutant SHIP2△935-1258 plasmids. Meanwhile, transwell migration and invasion assays indicated that the introduction of wild-type SHIP2 plasmids, but not deletion mutant SHIP2△935-1258 plasmids, inhibited the migration and invasion of GC cells (Figure 5C,D). Taken together, these results suggested that PRD and SAM domains of SHIP2 were crucial for the inhibitory effects of IQGAP2 on the migration and invasion of GC cells.

## 3. Discussion

In this report, we describe the role of IQGAP2, a novel SHIP2-interacting partner, in the migration and invasion of GC cells. Our data also demonstrate a novel role for C-terminal domains of SHIP2 in the regulation of its 5-phosphatase activity. Although the role of SHIP2 in tumor development and progression still remains controversial, our previous work [12] and the present study provide strong evidence that SHIP2 functions as a potential tumor suppressor in GC.

The human IQGAP family consists of three members, IQGAP1, IQGAP2 and IQGAP3, and each contains a calponin homology domain (CHD), WW domain, IQ-motif containing region (IQ), GAP-related domain (GRD), and Ras-GAP C-terminal domain (RGCT). The multiple domains in IQGAPs mediate protein–protein interactions with a number of binding partners which regulate diverse biology process, such as cytoskeleton remodeling, cell adhesion, Ca^2+^ and small G-protein signaling, protein trafficking, and tumorigenesis [29,30,31,32,33,34]. IQGAP1 is the best characterized and considered to promote tumorigenesis and progression, whereas IQGAP2 and IQGAP3 are less reported and their biological roles are poorly defined [35]. Recently, IQGAP2 has been considered as a tumor suppressor based on the observed reduction in IQGAP2 in hepatocellular carcinoma, prostate cancer, ovarian cancer and GC, which is further supported by the findings that IQGAP2 deficiency contributed to hepatocellular carcinoma tumorigenesis in mice and cell proliferation and invasion in hepatocellular carcinoma, prostate cancer, and ovarian cancer cell models [25,26,27,28,36]. However, in our study, there was no significant difference in IQGAP2 expression in GC cell lines and the normal gastric mucosal epithelial cell line. Moreover, the analyses of the TCGA database and HPA database for IQGAP2 expression in GC tissues showed consistent results with ours. In spite of the reduced mRNA expression of IQGAP2 from the analysis of Oncomine database, Dinesh Kumar et al. also showed that there was no significant change in its mRNA expression in any of the GC subtypes based on the TCGA database [37]. Indeed, other reports found that IQGAP2 was overexpressed in tissues of colon cancer and prostate cancer [38,39]. Despite the inconsistency, more thorough investigations are required to verify this postulation and ascertain whether it pertains only to limited tumors.

Although interaction between SHIP2 and cytoskeletal proteins, such as filamin, Vinexin, CAP, c-cbl, Arap3, p130Cas, RhoA etc, has been widely reported, it is not clear whether these interactions may influence SHIP2 phosphatase activity [40,41,42,43,44,45]. Here, we identified that a novel SHIP2-interacting protein, IQGAP2, could enhance SHIP2 enzyme activity through binding to its C-terminal PRD and SAM domains, whereas the binding of filamin and Vinexin to PRD, and the binding of RhoA to the N-terminal region between SH2 and catalytic domains, did not affect SHIP2 phosphatase activity [40,41,45]. Likewise, it has been reported that the insulin receptor tyrosine kinase substrate (IRTKS) interacted with SHIP2 by binding to its 5-phosphotase catalytic domain to suppress its enzyme activity, which further supports our findings in this work [46]. In addition, IQGAP2 has been found to promote E-cadherin expression and reduce EMT via inhibiting Akt activation [26]. However, how IQGAP2 inhibits Akt activation remains unclear. Our work first proposed the underlying mechanism that IQGAP2 interacted with SHIP2 and enhanced its phosphatase activity, thereby inactivating Akt and reducing EMT.

Though IQGAP2 methylation was reported to correlate with tumor invasion and poor prognosis of GCs [28], the role and underlying mechanism of IQGAP2 in GC cells migration and invasion remain to be elucidated. In this study, we found that, similar to SHIP2, the silencing of IQGAP2 promoted the migration and invasion of GC cells, which was associated with inhibiting SHIP2 phosphatase activity, activating Akt and subsequently increasing EMT. Next, to confirm that the interaction of IQGAP2 with SHIP2 is required for the inhibitory effect of IQGAP2 on GC cells migration and invasion, we knocked down IQGAP2 in ectopic SHIP2-overexpressing GC cells, and found that IQGAP2 knockdown reversed the inhibition of cell migration and invasion by SHIP2 induction. Furthermore, the deletion of PRD and SAM domains of SHIP2 abrogated the interaction and restored cell migration and invasion. These results clarify the inhibitory role of IQGAP2 in GC cells migration and invasion, and provide supporting evidence for its tumor-suppressive function.

Of note, as IQGAP2 and SHIP2 are scaffolding proteins, interaction between them may exist in an indirect manner. For instance, Rho GTPases Cdc42 and Rac1 interact with IQGAP2 by binding to GRD of IQGAP2, which influences actin polymerization [47]. Besides, RhoA has been reported to interact with SHIP2 to regulate cell polarization and migration [45]. This suggests that IQGAP2, SHIP2, and Rho GTPases may be assembled as a large scaffolding complex to regulate cytoskeleton dynamics, therefore affecting cell motility and invasion. In addition to the effect on cell migration and invasion, IQGAP2 and SHIP2 also mediate metabolism and insulin sensitivity [48,49], which implies that their interaction may participate in these processes and render an interesting issue for future study.

## 4. Materials and Methods

### 4.1. Cell lines and Cell Culture

The human GC cell lines and HEK293T cell line were purchased from the Institute of Biochemistry and Cell Biology, Shanghai Institutes for Biological Sciences, Chinese Academy of Sciences. The normal gastric mucosal epithelial cell line GES-1 was obtained from Cancer Institute & Hospital, Chinese Academy of Medical Sciences. All these cells were grown in Dulbecco’s modified Eagle’s medium (DMEM) supplemented with 10% fetal bovine serum (FBS), 1% Penicillin-Streptomycin Solution at 37 °C, 5% CO^2^.

### 4.2. Plasmids and Lentiviruses

The pCMV-AC-GFP-SHIP2 constructs (tGFP-SHIP2^WT^) were purchased from Origene (RG214716). The truncated fragments of SHIP2 were sub-cloned into pCMV-AC-GFP vectors. The pCDNA3.1-cMyc-IQGAP2 constructs were obtained from Sangon Biotech (Shanghai, China). Cells were transfected with 2 µg plasmids or empty vectors in Opti-MEM medium (Invitrogen, Carlsbad, CA, USA) using Lipofectamine 2000 reagent (Invitrogen, Carlsbad, CA, USA) according to the manufacturer’s protocol.

For lentivirus-mediated SHIP2 and IQGAP2 short hairpin RNA (shRNA) silencing, specific shRNA sequences (SHIP2-shRNA: 5’-GTCCATGGATGGCTATGAA-3’, IQGAP2-shRNA: 5’-GCTCCTACCTACTGCGAAT-3’) were cloned into the lentiviral vector GV248 (GeneChem, Shanghai, China). The virus was produced, and target cells were infected according to the manufacturer’s protocol. SGC-7901 and MGC-803 cells were infected by lentiviruses carrying vector silencing either SHIP2 or IQGAP2, and GFP-positive cells were selected by fluorescence-activated cell sorting (FACS) to generate stable clones of SGC-7901 cells (SGC-7901.sh-SHIP2, SGC-7901.sh-IQGAP2 and SGC-7901.sh-scramble) and MGC-803 cells (MGC-803.sh-SHIP2, MGC-803.sh-IQGAP2 and MGC-803.sh-scramble).

### 4.3. Immunoprecipitation Assay

Cells were incubated with immunoprecipitation lysis buffer for 30 min on ice. The cell lysates (1 mg total protein) were incubated with the indicated primary antibodies (3 μg) at 4 °C overnight. Then, protein A/G beads (Santa Cruz, Santa Cruz, CA, USA) were added to cell lysates, and incubated at 4 °C for 2 h. After washing the beads, protein samples were analysed by Western blot analysis.

### 4.4. Western Blot Analysis

Cultured cells were lysed in an RIPA lysis buffer. Protein samples were separated by SDS-polyacrylamide gel electrophoresis and then transferred to Nitrocellulose Membranes (Millipore, Burlington, MA, USA), which were subsequently incubated with primary antibodies and secondary antibodies. The primary antibodies used were as follows: anti-SHIP2 rabbit monoclonal antibodies (1:1000, Abcam, Cambridge, UK); anti-IQGAP2 mouse monoclonal antibodies (1:500, Santa Cruz, Santa Cruz, CA, USA); anti-GAPDH mouse monoclonal antibodies (1:1000, Zhongshan Golden Bridge Biotechnology, Beijing, China); anti-pAkt and anti-Akt rabbit polyclonal antibodies (1:1000, Cell Signaling Technology, Beverly, MA, USA); anti-E-Cadherin, anti-β-Catenin, anti-N-Cadherin rabbit monoclonal antibodies (1:1000, Cell Signaling Technology, Beverly, MA, USA); anti-tGFP mouse monoclonal antibodies (1:2000, Origene, Rockville, MD, USA); anti-cMyc mouse monoclonal antibodies (1:1000, Cell Signaling Technology, Beverly, MA, USA). Blots were visualized by Tanon 4500SF image system (Tanon, Shanghai, China). 

### 4.5. Immunofluorescence Staining

MGC-803 cells were fixed with 4% paraformaldehyde for 10 min at room temperature and then treated with phosphate-buffered saline (PBS) containing 0.1% Triton X-100. After blocking with goat serum for 30 min at room temperature, the cells were incubated with anti-SHIP2 rabbit antibodies (1:50, Abcam, Cambridge, UK) and anti-IQGAP2 mouse antibodies (1:50, Santa Cruz, Santa Cruz, CA, USA) at 4 °C overnight. After incubation with Alexa Flour 488 anti-rabbit IgG antibodies and Alexa Flour 594 anti-mouse IgG antibodies (1:200, Proteintech, Wuhan, China) for 2 h and then nuclei staining with 4’, 6-diamidino-2-phenylindole (DAPI, Beyotime Biotechnology, Shanghai, China), specimens were observed under a confocal laser scanning microscope (Carl Zeiss, Oberkochene, Germany).

### 4.6. Wound Healing and Transwell Assays

For wound-healing assays, 4 × 105 cells were seeded in each well and a line was drawn with a 200 μL pipette tip on the next day. After washing with PBS twice, serum-free media was re-added and cells continued to grow for 24 and 48 h. Microphotographs were taken by phase-contrast microscope (Olympus, Japan).

Cell invasion and migration assays were detected by Transwell assays with or without BD matrigel. 5 × 104 cells in 200 μL of serum-free DMEM were seeded into the upper chambers, and DMEM medium with 10% fetal bovine serum (FBS) was added to the lower chambers. The cells invaded through matrigel for 48 h (invasion assays). Cells that successfully invaded to the lower surface of the membrane were fixed with methanol and stained with 0.5% crystal violet solution, then were photographed and counted by inverted phase contrast microscope (Olympus, Japan).

### 4.7. RNA Isolation and Real-Time Quantitative PCR Analysis (qRT-PCR)

Total RNA was extracted by TRIzol Reagent (Invitrogen). RNA samples were reverse transcribed with 5 × HiScript®II qRT SuperMix. cDNA was then amplified with specific primers and Power SYBR Green PCR Master Mix (Applied Biosystems). GAPDH mRNA levels were as internal control. The specific primers of SHIP2 were as follows: 5’-CGAGAACCGTATCAGCCATGTCAG-3’ (sense), 5’-GCAGCCGCAGGATGTCCAAG-3’ (antisense); for IQGAP2, 5’-ACATGGTGAGCCGTGCAATGATAG-3’ (sense), 5’-CCAGTCTGTTCCAACGTCCGAAG-3’ (antisense).

### 4.8. Malachite Green Phosphatase Assays

Malachite green phosphatase assays were performed using Malachite Green Phosphate Assay Kit (Sigma-Aldrich, St. Louis, MO, USA) according to the manufacturer’s instructions. Briefly, SHIP2 was immunoprecipitated from the lysates of GC cells with the indicated treatment. Beads were washed four times with lysis buffer. A total of 80 μL of protein samples were transferred into separate wells of the 96-well plate. Then, 20 μL of Working Reagent was added to each well. Samples are incubated for 30 min at room temperature. Absorbance at 630 nm was measured to calculate free phosphate concentration, which was used to evaluate enzyme activity.

### 4.9. Quantitation and Statistical Analysis

Western blot and wound-healing assays were quantitated with Image J software version 1.8.0 (National Institutes of Health, Bethesda, MD, USA). Statistical analyses were performed by GraphPad Prism version 6.0 (GraphPad Software, San Diego, CA, USA). Paired Student *t* tests were used to compare the differences between two groups. All quantitative data were expressed as mean ± SEM, and the data were obtained from at least three independent experiments. *p*-values less than 0.05 were considered statistically significant.

## Figures and Tables

**Figure 1 ijms-21-01968-f001:**
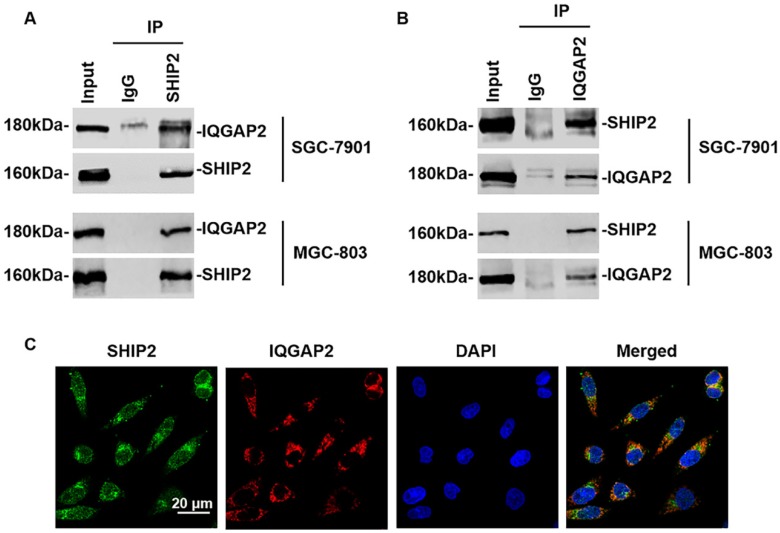
SHIP2 physically interacts with IQGAP2 in GC cells. (**A**,**B**) Interaction between SHIP2 and IQGAP2 was detected by co-immunoprecipitation (co-IP) with anti-SHIP2 or anti-IQGAP2 antibodies in SGC-7901 and MGC-803. The Immunoglobulin G (IgG) group acted as negative control; (**C**) Co-localization of SHIP2 and IQGAP2. MGC-803 cells were immunostained with anti-SHIP2 and anti-IQGAP2 antibodies followed by incubation with Alexa-488 conjugated anti-rabbit antibodies and Alexa-594 conjugated anti-mouse antibodies, respectively. Images were taken with a confocal microscope. Data shown are representative of three individual experiments.

**Figure 2 ijms-21-01968-f002:**
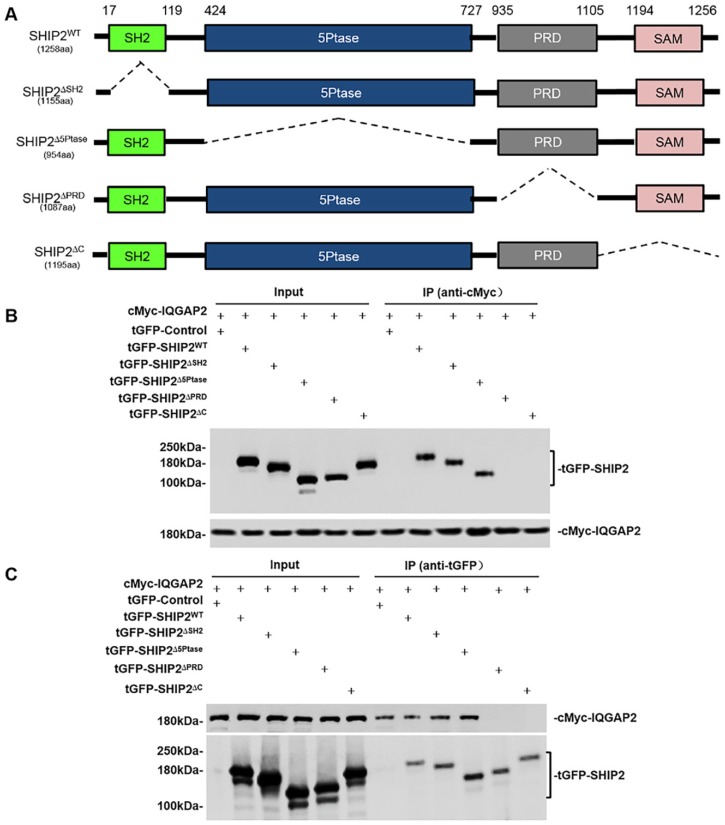
IQGAP2 binds to PRD and SAM domains of SHIP2. (**A**) A schematic diagram of tGFP-tagged SHIP2 constructs encoding its full length and different deletion domains; (**B**,**C**) A series of SHIP2 constructs and cMyc-IQGAP2 plasmids were co-transfected into HEK293T cells. The cell lysates were immunoprecipitated with anti-cMyc antibodies or anti-tGFP antibodies, and then subjected to Western blot analysis. Data shown are representative of three individual experiments.

**Figure 3 ijms-21-01968-f003:**
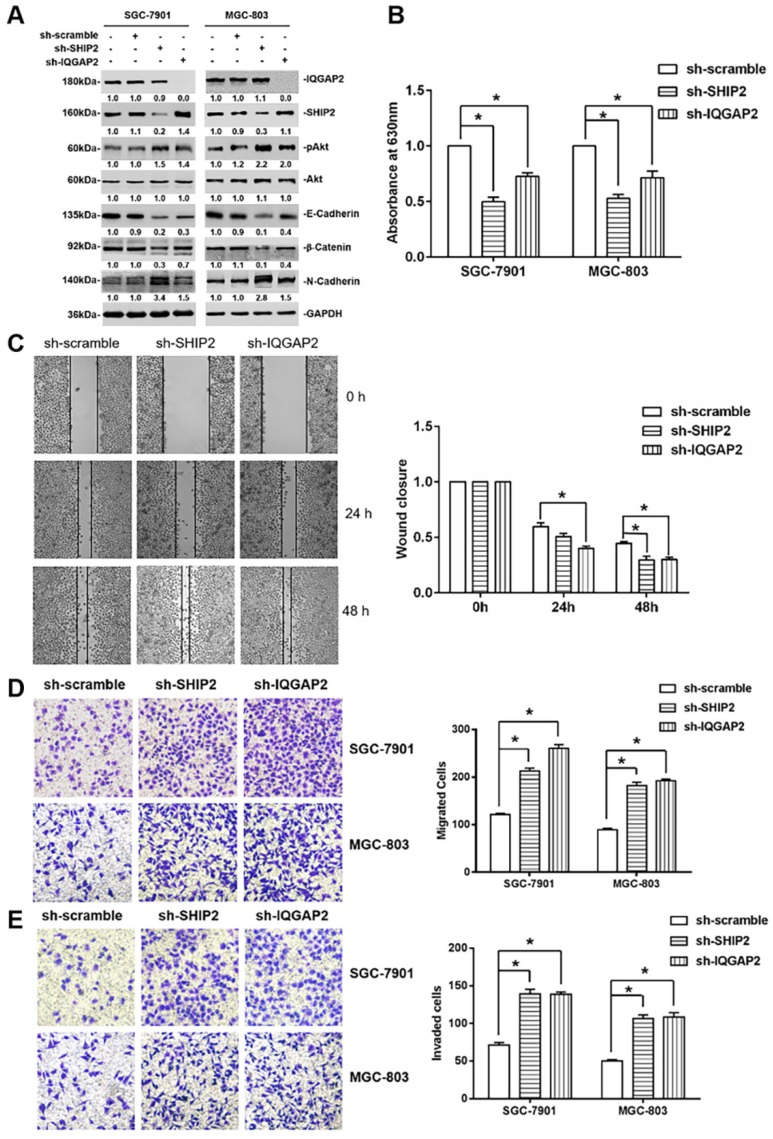
IQGAP2 inhibits the migration and invasion of GC cells in association with SHIP2 phosphatase activity. (**A**) Whole-cell lysates from stable SGC-7901 and MGC-803 cells carrying a lentivirus-mediated IQGAP2 or SHIP2 shRNA, and their corresponding counterparts were subjected to Western blot analysis for the indicated proteins. Data shown are representative of three individual experiments; (**B**) SHIP2 proteins, immunoprecipitated by anti-SHIP2 antibodies in stable SGC-7901 and MGC-803 cells carrying a lentivirus-mediated IQGAP2 or SHIP2 shRNA, were used to detect its phosphatase activity by malachite green phosphatase assays (*n* = 3, mean ± SEM, * *p* < 0.05); (**C**) the migration of SGC-7901 cells carrying a lentivirus-mediated IQGAP2 or SHIP2 shRNA was detected by wound-healing assays. Right panel shows quantitation of wound closure corresponding to the left panel. The y axis represents the percentages of wound closure at 24 or 48 h after wound introduction (*n* = 3, mean ± SEM, * *p* < 0.05); (**D**,**E**) Migration and invasion of SGC-7901 and BGC-823 cells carrying a lentivirus-mediated IQGAP2 or SHIP2 shRNA were detected by Transwell assays (*n* = 3, mean ± SEM, * *p* < 0.05). Magnification, ×200.

**Figure 4 ijms-21-01968-f004:**
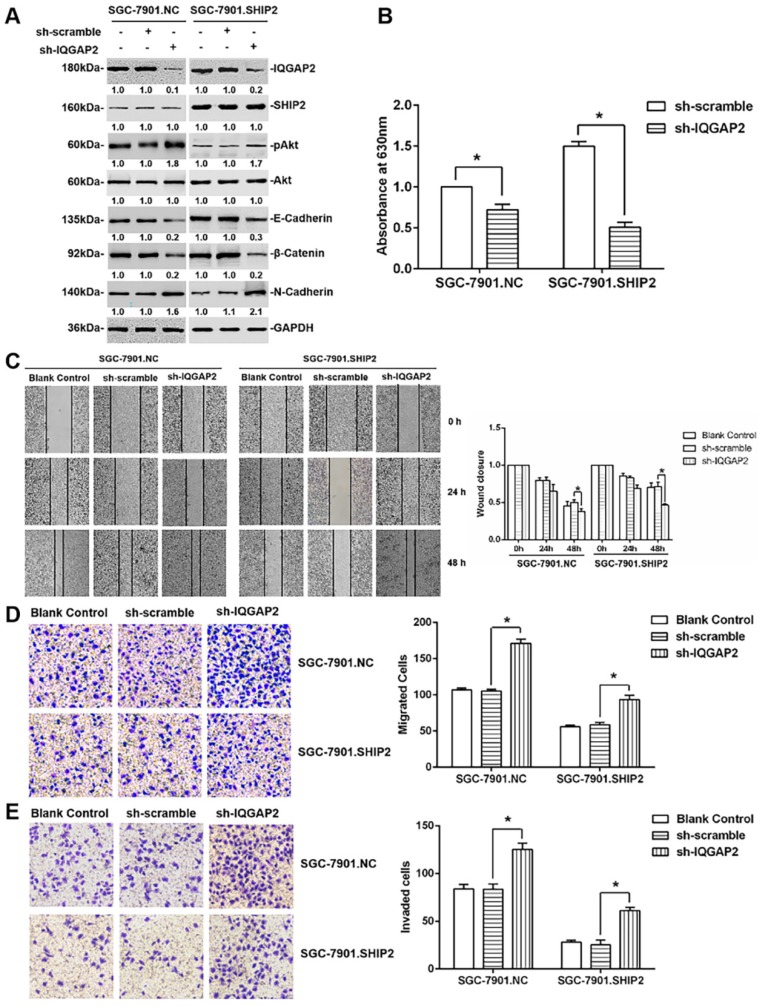
The inhibition of IQGAP2 on the migration and invasion of GC cells is partially due to elevating SHIP2 phosphatase activity. (**A**) Whole-cell lysates from stable SGC-7901.SHIP2 and SGC-7901.NC cells transduced with IQGAP2 or scramble shRNA were subjected to Western blot analysis for the indicated proteins. Data shown are representative of three individual experiments; (**B**) SHIP2 proteins, immunoprecipitated with anti-SHIP2 antibodies in stable SGC-7901.SHIP2 and SGC-7901.NC cells transduced with IQGAP2 or scramble shRNA, were used to detect its phosphatase activity by malachite green phosphatase assays (*n* = 3, mean ± SEM, * *p* < 0.05); (**C**) the migration of stable SGC-7901.SHIP2 and SGC-7901.NC cells transduced with IQGAP2 or scramble shRNA was detected by wound-healing assays. Right panel shows quantitation of wound closure corresponding to the left panel. The y axis represents the percentages of wound closure at 24 or 48 h after wound introduction (*n* = 3, mean ± SEM, * *p* < 0.05); (**D,E**) Migration and invasion of stable SGC-7901.SHIP2 and SGC-7901.NC cells transduced with IQGAP2 or scramble shRNA were detected by Transwell assays (*n* = 3, mean ± SEM, * *p* < 0.05). Magnification, ×200.

**Figure 5 ijms-21-01968-f005:**
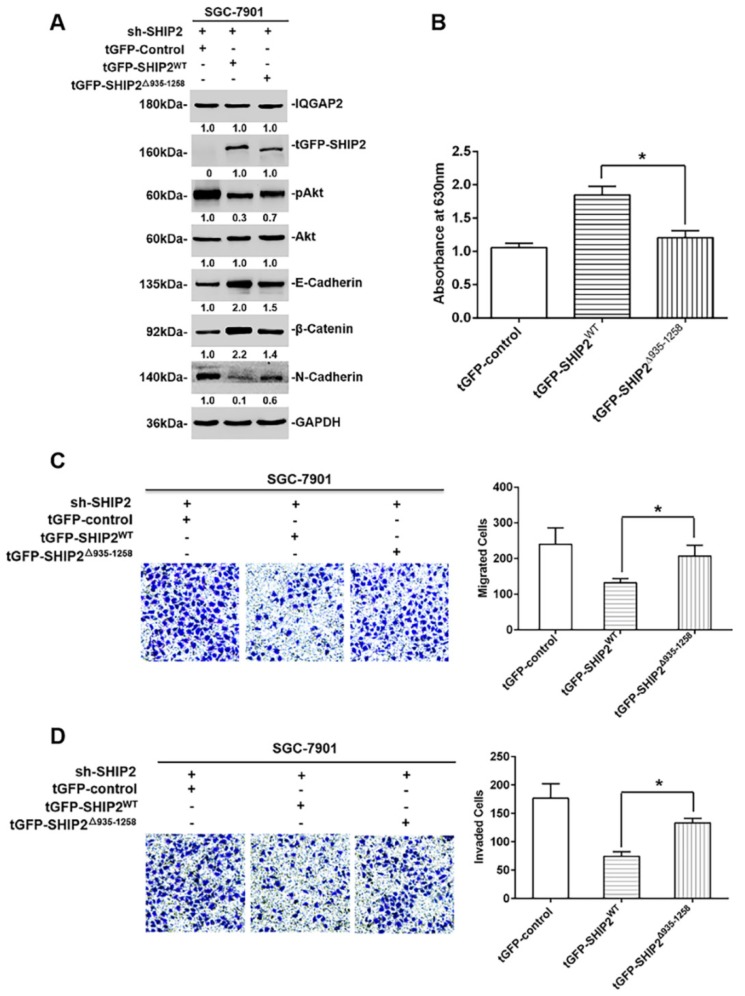
Binding to PRD and SAM domains of SHIP2 is crucial for the inhibitory effects of IQGAP2 on the migration and invasion of GC cells. (**A**) Whole-cell lysates from stable SGC-7901.shSHIP2 cells transfected with wild-type SHIP2 and deletion mutant SHIP2Δ935-1258 plasmids were subjected to Western blot analysis for the indicated proteins. Data shown are representative of three individual experiments; (**B**) SHIP2 proteins, immunoprecipitated by anti-SHIP2 antibodies in stable SGC-7901.shSHIP2 cells (transfected with wild type SHIP2 and deletion mutant SHIP2Δ935-1258 plasmids), were used to detect its phosphatase activity by malachite green phosphatase assays (*n* = 3, mean ± SEM, * *p* < 0.05); (**C**,**D**) the migration and invasion of stable SGC-7901.shSHIP2 cells transfected with wild-type SHIP2 and deletion mutant SHIP2Δ935-1258 plasmids were detected by Transwell assays (*n* = 3, mean ± SEM, * *p* < 0.05). Magnification, ×200.

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
