# Peer review of "IQGAP2 Inhibits Migration and Invasion of Gastric Cancer Cells via Elevating SHIP2 Phosphatase Activity"

_ijms, 2020, doi:10.3390/ijms21061968_

Round 1

Reviewer 1 Report

The manuscript presents a mechanistic investigation of the functional interaction of SHIP2 and IQGAP2 in the cellular migration and invasion in cancer. The investigation of the mechanism, and going beyond a presentation of results is a welcome aspect of the manuscript.

I have a few minor points that will help in the paper's presentation:

  • Molecular weights should be annotated with the corresponding Western blot bands in the figures.
  • Are antibodies in Figs 2&3 the same? If so, why call them anti-SHIP2 in Fig 2, and anti-GFP in Fig 3. If not, why use different antibodies? Section 4.4 doesn’t talk about anti-GFP or ant-Myc antibodies. SHIP2 domains were ablated in Fig 3, so perhaps anti-SHIP2 antibodies were replaced by anti-GFP antibodies, but why not use just anti-IQGAP2 antibodies?
  • Fig 4A. downregulation of beta-catenin is arguable.
  • 4C can’t be the % wound closure since it is increasing with time. Also, the max is somehow fixed at 1, rather than 100 (if it was %).
  • Fig 6 hints toward the role of SAM and PRD domain binding, but all that it proves is that SAM and PRD domains have a role in the invasive, migratory or EMT roles of SHIP2. The SAM and PRD domains could be acting in a mechanism independent of IQGAP2 binding.
  • Fig 4A shows that shIQGAP2 increases the expression of SHIP2, moreso in SGC-7901. Line 10-11 of page 5 suggests the opposite.

Is it possible to ablate the phosphorylation domain of SHIP2 and observe if the effect of IQGAP2 is ablated? While the data suggest that the effect of IQGAP2 is mediated by SHIP2 activity, I don't think it definitely proves it.

Author Response

Response to Reviewer 1 Comments

Point 1: Molecular weights should be annotated with the corresponding Western blot bands in the figures.

Response 1: As suggested, molecular weights for the corresponding Western blot bands have been annotated in the figures.

Point 2: Are antibodies in Figs 2&3 the same? If so, why call them anti-SHIP2 in Fig 2, and anti-GFP in Fig 3. If not, why use different antibodies? Section 4.4 doesn’t talk about anti-GFP or ant-Myc antibodies. SHIP2 domains were ablated in Fig 3, so perhaps anti-SHIP2 antibodies were replaced by anti-GFP antibodies, but why not use just anti-IQGAP2 antibodies?

Response 2: In Figure 2 (re-ordered as Figure 1 in revised manuscript), the specific antibodies against SHIP2 were used to detect the endogenous expression of SHIP2 in co-IP assays. But in Figure 3(re-ordered as Figure 2 in revised manuscript), the tag antibodies such as anti-tGFP and anti-cMyc, were used to detect the exogenous expression of SHIP2 and IQGAP2 in co-IP assays. In this experiment, wild type and several truncated fragments of SHIP2 were sub-cloned into pCMV-AC-tGFP vectors, and the specific antibodies against SHIP2 could not recognize all SHIP2 truncates. Therefore, we used anti-tGFP tag antibodies to perform this experiment. In fact, anti-cMyc tag and anti-IQGAP2 antibodies were used to precipitate and detect IQGAP2 in co-IP assays and the results were consistent, so we just showed one of the results. In addition, the information of tag antibodies has been added in Section 4.4.

Point 3: Fig 4A. downregulation of beta-catenin is arguable.

Response 3: We have quantitated the relative grey values of Western blot bands and marked at the bottom of each band. In Figure 4A(re-ordered as Figure 3A in revised manuscript), although the change of beta-catenin is not very obvious in SGC-7901 cell lines, beta-catenin is downregulated in MGC-803 cell lines.

Point 4: Fig 4C can’t be the % wound closure since it is increasing with time. Also, the max is somehow fixed at 1, rather than 100 (if it was %).

Response 4: As suggested, we have re-marked y-axis in Figures 4C and 5C(re-ordered as Figures 3C and 5C in revised manuscript).

Point 5: Fig 6 hints toward the role of SAM and PRD domain binding, but all that it proves is that SAM and PRD domains have a role in the invasive, migratory or EMT roles of SHIP2. The SAM and PRD domains could be acting in a mechanism independent of IQGAP2 binding.

Response 5: Actually, Figure 6(re-ordered as Figure 5 in revised manuscript) is the continued results from Figure 5(re-ordered as Figure 4 in revised manuscript). We just verified IQGAP2 plays an inhibitory role in cell migration and invasion by binding to PRD and SAM domains of SHIP2, thereby elevating SHIP2 phosphatase activity. Therefore, the results of Figure 6(re-ordered as Figure 5 in revised manuscript) stressed the important role of PRD and SAM domains involved in the interaction.

Point 6: Fig 4A shows that shIQGAP2 increases the expression of SHIP2, moreso in SGC-7901. Line 10-11 of page 5 suggests the opposite.

Response 6: As mentioned in Response 3, the quantitated SHIP2 bands have been marked at the bottom of each band. In SGC-7901 cell lines, knockdown of IQGAP2 just had a slight change of SHIP2 expression, whereas no significant change of that in MGC-803 cell lines. We repeated this experiment more than three times, and could not make the significant conclusions.

Point 7: Is it possible to ablate the phosphorylation domain of SHIP2 and observe if the effect of IQGAP2 is ablated? While the data suggest that the effect of IQGAP2 is mediated by SHIP2 activity, I don't think it definitely proves it.

Response 7: I think it is a constructive suggestion. Although we focused on the effect of the interaction of SHIP2 and IQGAP2 on SHIP2 phosphatase activity, deletion of the key 5-phosphatase domain of SHIP2 is a direct way to confirm this effect. In the follow-up work, we intend to perform this experiment.

Reviewer 2 Report

The study describes a role of IQGAP2 and SHIP2 in GC cell migration. IQGAP2 is required for SHIP2 activity (via association of SHIP2 through its C-terminus to IQGAP2) and this promotes phosphatase activity, down regulation of Akt and migration. The data are mostly sound and novel.

Figs 4C and 5C and not convincing. The slit width varies making it hard to assess the degree of wound healing.

The changes in cadherin expression are not sufficient to claim EMT.

Title of Fig 5 is misleading because knockdown was used, not overexpression.

The text presentation requires editing. Numerous mistakes in grammar are found.

Fig 1 could be made a supplemental figure.

Author Response

Response to Reviewer 2 Comments

Point 1: Figs 4C and 5C are not convincing. The slit width varies making it hard to assess the degree of wound healing.

Response 1: For wound healing assays, to assess the degree of wound healing by use of calculating the slit width varies is an ordinary and accepted method. In this manuscript, we quantitated the slit width of wound healing assays and showed quantitation of wound closure in Figures 4C and 5C (re-ordered as Figures 3C and 4C in revised manuscript). The statistical results were significant. Additionally, transwell migration assays in Figures 4D and 5D(re-ordered as Figures 3D and 4D in revised manuscript) also confirmed the inhibitory effect of the interaction of SHIP2 and IQGAP2.

Point 2: The changes in cadherin expression are not sufficient to claim EMT.

Response 2: We have quantitated the relative grey values of Western blot bands and marked at the bottom of each band. In Figures 4A, 5A, and 6A(re-ordered as Figures 3A, 4A, and 5A in revised manuscript), there was a slight variation of E-cadherin and N-cadherin expression, which implied EMT development.

Point 3: Title of Fig 5 is misleading because knockdown was used, not overexpression.

Response 3: Because of the relative high expression of IQGAP2 in GC cells, we validated its effect on migration and invasion by use of knockdown IQGAP2. Title of Figure 5 (re-ordered as Figure 4 in revised manuscript) just describes the function of IQGAP2, which is consistent with the results of this experiment.

Point 4: The text presentation requires editing. Numerous mistakes in grammar are found.

Response 4: We asked a native English scholar to check the English language and style of the revised manuscript.

Point 5: Fig 1 could be made a supplemental figure.

Response 5: As suggested, we re-ordered the figures in the manuscript and moved Figure 1 into supplementary materials.